# Forecasting for Haditha reservoir inflow in the West of Iraq using Support Vector Machine (SVM)

Othman A. Mahmood[1], Sadeq Oleiwi Sulaiman[1], Dhiya Al-Jumeily[2]*

1 Dams and Water Resources Engineering Department, College Engineering, University of Anbar, Anbar, Iraq, 2 Faculty of Engineering and Technology, Liverpool John Moores University, Liverpool, British

* d.aljumeily@ljmu.ac.uk

**Data Availability Statement:** All relevant data are within the manuscript and its Supporting Information files.

**Funding:** The author(s) received no specific funding for this work.

## Abstract

Accurate inflow forecasting is an essential non-engineering strategy to guarantee flood management and boost the effectiveness of the water supply. As inflow is the primary reservoir input, precise inflow forecasting may also offer appropriate reservoir design and management assistance. This study aims to generalize the machine learning model using the support vector machine (SVM), which is support vector regression (SVR), to predict the discharges of the Euphrates River upstream of the Haditha Dam reservoir in Anbar province West of Iraq. Time series data were collected for the period (1986-2024) for the river's daily, monthly, and seasonal flow. Different kernel functions of SVR were applied in this study. The kernels are linear, Quadratic, and Gaussian (RBF). The results showed that the daily time scale is better than the monthly and seasonal performance. In contrast, the linear kernel outperformed the other SVR kernel with a time delay of one day based on the value of the coefficient of determination ($R^2$ = 0.95) and the root mean square error (RMSE = 53.29) $m^3$/sec for predicting daily river flow. The results showed that the proposed machine learning model performed well in predicting the daily flow of the Euphrates River upstream of the Haditha Dam reservoir; this indicates that the model might effectively forecast flows, which helps improve water resource management and dam operations.

## Introduction

Water flow is the basis of life on Earth and is considered one of the basic needs of living and plant organisms, as well as the prosperity of agriculture and ensuring survival. Rivers are considered one of the primary sources of water on the surface of the Earth. Therefore, it has become necessary to study the quantity of river flow, as it is one of the main factors for achieving sustainable development of water resources issues. These studies include the design of hydraulic facilities and proper planning of water projects, in addition to operating water reservoirs, flood control, and treating droughts. It must be noted that the economic return of any region is directly related to the quantity and quality of water sources in that region. Therefore, we must work to protect and manage water resources well, develop sustainable strategies to

**Competing interests:** The authors have declared that no competing interests exist.

**Abbreviations:** $Q_{obs}$, *Value of the observation discharge data*, $m^3/sec$; $Q_{pre}$, *Value of the predicted discharge data*, $m^3/sec$; Qobs−, *Mean values of observation discharge data*, $m^3/sec$; Qobs−, *Mean values of observation discharge data*, $m^3/sec$; Qpre−, *Mean values of prediction discharge data*, $m^3/sec$; n, *Number of real data*; b, *Bias*; C, *Regularization parameter*; f, *Function of*; J, *Lagrange functional*; K, *kernel function*; L, *Insensitive loss function*; N, *Number of input examples*; t, *Time step*; W, *Weight vector*; $W^T$, *Transpose of vector w*; T, *Transpose of*; X, *Input vector of independent variables, and*; ||X||, *Euclidean norm length of vector x*; α, *Lagrange multiplier SVM parameter*; ε, *Error insensitive zone prescribed parameter*; ξ, *Constraint violation estimation error*; σ, *Width of the radial basis kernel function, and*; ϕ, *Function to be minimized.*

provide clean and sustainable water for all and balance the needs of humans and aquatic ecosystems. Recent studies indicate that river flow levels are becoming unstable and are at increasing risk of drying out. Rivers in Iraq, in general, are vulnerable to this risk for several reasons, including climate change and human use of water. Therefore, many studies have been conducted to provide sufficient data to estimate river flow amounts and predict drought. Many hydrological models have been developed to analyze and predict river water flow, giving practical tools for water resource management and sustainable decision-making. These hydrological models are based on a wide range of data, including climate information, past water information, and hydrological system behavior. These models use many equations and variables to estimate water flow and predict drought and are valuable tools for hydrological planning and water resource management under changing conditions. Using these models and sufficient data, those interested in water resources management can analyze and evaluate drought risk and take the necessary measures to deal with it, contributing to the sustainable development of water resources issues. Forecasting future river flows is essential for making decisions in water resource management and water project planning. Future forecasting depends on analyzing past and current variables that describe the hydrological phenomenon to be studied. Accurate river flow prediction has been a significant challenge in flood management and reducing damage and potential threats to life. Accordingly, it is increasingly important to use reliable river flow forecasting methods to enable timely and effective planning of water resource use [1]. Accurate river flow prediction can be essential in water resource planning and management. However, many complex factors influence this phenomenon, making it challenging to analyze [2]. Therefore, it becomes necessary to incorporate the influencing factors into a model that can estimate river flow with acceptable accuracy [3]. Currently, intelligent systems are widely used to estimate nonlinear phenomena, and one of the methods used in hydrology is the support vector machine model (SVM). The support vector machine (SVM) model is effective, and optimization algorithms have been developed in recent years to improve its performance, increase its accuracy, and reduce the error rate in river flow prediction. Accurate river flow forecasting can be achieved using the SVM model, and this information can be used in planning and water resource management processes. The SVM model can be applied in hydrological analyses to improve river flow prediction and achieve more accurate results. It is important to note that using advanced models such as SVM requires consideration of optimizing algorithms and tuning parameters and providing sufficient and appropriate training data to achieve the best model performance in river flow prediction. Forecasting river discharges enhances sustainability, reduces the negative impacts of floods and droughts, and improves water resources management and environmental planning. It also supports strategic decision-making in risk management and planning for the future, enhancing water resources' sustainability and balancing the diverse water needs of different communities and sectors [4]. Many researchers have used artificial intelligence techniques, including support vector machines, to predict future river discharges in other regions. A study used a support vector machine (SVM) model to forecast monthly flow at the Huaxian station in China. The study found that the proposed SVM model accurately predicted monthly flow at the station. The SVM model was trained to estimate future flow patterns by analyzing and utilizing historical monthly flow data. The results demonstrated that the SVM model successfully achieved accurate predictions of monthly flow at the Huaxian station; this indicates that the SVM model can be an effective water resource planning and management tool in the specified area [5]. In a study, artificial neural networks (ANNs) and support vector machine (SVM) models in forecasting storm water runoff in the Roodak watershed northeast of Tehran. Was used 92 Modis sensors to collect data during the statistical period from 2003 to 2005. According to the study, the SVM model showed acceptable performance in estimating rainwater runoff in the

mentioned area; this means that the model could predict the amount of water flowing on the surface based on the sensed data [6]. Also, a study compared support vector machines (SVM) and artificial neural networks to predict daily flow in the Cypress River in Texas. The results of the study found that the SVM model outperformed the neural networks, demonstrating superior accuracy. This highlights the effectiveness of SVM in analyzing data and predicting hydrological models [7]. In a study, a support vector machine (SVM) was used to forecast monthly flow. The SVM model's performance was enhanced by employing principal components analysis (PCA) for preprocessing the input variables. The study demonstrated that incorporating PCA as an optimization technique improved the SVM model's accuracy in predicting monthly flow; this highlights PCA's positive impact on enhancing prediction accuracy and the model's effectiveness in handling data [8]. A study analyzed the Wei River in China to forecast 10-day inflows using input factors like inflow, precipitation, relative humidity, minimum and maximum temperature, and precipitation projections. The three methods used in the study were compared using the available data: Artificial Neural Networks (ANN), Support Vector Regression (SVR), and Adaptive Neuro-Fuzzy Inference System (ANFIS). The results indicated that changes in income variables significantly impacted forecast uncertainty. The contribution of data-driven models was limited and varied seasonally, performing better in winter and summer but needing to be more critical in spring and fall [9]. A study was conducted on the Three Gorges Reservoir; the predicted monthly flow using three machine learning models: genetic programming (GP), seasonal autoregressive (SAR), and autoregressive neural kernel (SVR). The RBF was used as the kernel Influential in the SVR model. The results showed that the performance of the SVR and GP models improves when coupled with independent principal components analysis (SSA) for flow series forecasting [10]. A study conducted in the Sutami Watershed in Indonesia used a Wavelet Support Vector Machine (WSVM) with an adapted RBF kernel to predict flow in the reservoir. WSVM is a machine learning model based on wavelet support, a technique used for classification and prediction. The RBF (Radial Basis Function) kernel has been adapted in the model to improve its performance in flow prediction. The results showed that the WSVM model better predicted the inflow into the reservoir using the RBF kernel [11]. In a study, a support vector machine (SVM) model was used to forecast lake water levels. It compared with a multilayer perceptual (MLP) model and a multiplicative seasonal autoregressive (SAR) model. The results of the SVM forecast were found to be more accurate than the other two models in predicting lake water levels for several months. These results suggest that support vector machine (SVM) models can be effective in predicting lake water levels [12]. A study used the Muskingum model to predict floods in the United States of America and the United Kingdom. A combination of a hybrid of the bat algorithm (BA) and the particle swarm optimization (PSO) algorithm, i.e., the hybrid bat-swarm algorithm (HBSA) was used. The results showed that the Muskingum model represented by (HBSA) achieved excellent performance compared to other methods based on the squared deviations (SSD), the sum of the absolute deviations (SAD), the peak discharge error, and the time-to-peak error [13]. A study used the Adaptive Neuro-Fuzzy Inference System (ANFIS) model to study the case of the influence of climate on monthly flow in the Aydoughmoush basin in Iran for the period 1987 - 2007. The bat algorithm (BA), particle swarm optimization (PSO), and Genetic Algorithm (GA) were used to obtain the ANFIS parameter and obtain the best ANFIS structure. The results showed better climate index performance with six months' delays. The study indicated that ANFIS-BA obtained better results than ANFIS-PSO and ANFIS-GA, with a root mean square error (RMSE) of 25% and 30% less than ANFIS-PSO and ANFIS-GA, respectively [14]. A study used the Support Vector Machine Method (SVM) at meteorological stations in Mosul and Baghdad. It analyzed different weather variables and found the SVM method successfully predicted wind speed, rainfall amounts, and humidity at the Mosul station

($R^2$ = 0.92). These variables can improve the accuracy of weather forecasts in the region [15]. In a study, was used two artificial intelligence models to predict subsurface evaporation rates, represented by the generalized neural network model for regression and the neural network for the radiative basis function. The model's input variables for this model include temperature, wind speed, humidity, and water depth. The applied models utilize actual hydrological and climatological in an arid region in the Iraqi Western Desert for two soil types. The results showed that the neural network model (ANN) could accurately predict subsurface reservoir evaporation based on the correlation coefficient, which reached (0.936) for fine gravel soil and (0.959) for coarse gravel soil [16]. A study used a deep learning model to predict river courses on data from the Tigris River in Iraq. Two methods were used to collect samples: linear deep learning (LDL) and stratified deep learning (SDL) in deep learning algorithms. The results indicated that stratified deep learning (SDL) improves accuracy by approximately 7.96-94.6 concerning several evaluation criteria. Thus, it is worth noting that SDL outperforms (LDL) in monthly streamflow modelling [17].

This study aims to use artificial intelligence with an SVM model to predict the discharge of the Euphrates River upstream of the Haditha Dam and improve water resources management and dam operations. The study uses historical data on the discharge of the Euphrates River upstream of Haditha Dam and information about flow behavior and the impact of the recently constructed dams in upstream countries. Also, this study aims to analyze and evaluate the relationship between the river's daily, monthly, and seasonal discharges and the effect of previous values on predicting and better understanding the behavior of the water drainage system.

## Materials and methods

### Study area

The Euphrates River is considered the primary water source in Anbar, Iraq, as most of the governorate's cities are located on its banks and depend on the river's water for their municipal, industrial, and agricultural needs [18]. The Euphrates River is an international river that passes through Turkey, Syria, and Iraq. The length of the Euphrates River in the governorate is about 450 km, representing 43% of the total length inside Iraq (1,160 km) and 17% of the entire length of the river from its source in Turkey to the mouth of the river. The Euphrates River enters Iraqi territory at Al-Qaim in Anbar Governorate. It constitutes a vital artery for the governorate's economic, industrial, and agricultural life. The Euphrates River in Anbar province feeds many agricultural areas, contributes to producing essential crops such as wheat, barley, and corn, and provides drinking water for the governorate's residents [19]. Therefore, maintaining regular flow levels in the Euphrates River downstream of Haditha Dam is essential to ensuring Iraq's future development. The Euphrates River and the Haditha Dam reservoir are located between latitudes (34° 40' and 34° 13') north and longitudes (42° 26' and 41° 55') east. The highest flood water level for the reservoir and the river is at level 147 and covers about 500 km$^2$ with 10 km of shoreline [20]. Fig 1 shows the Euphrates River upstream of the Haditha Dam reservoir in Anbar Governorate, western Iraq [21]. The Euphrates River in Anbar faces many challenges. The biggest challenge is the continuous decline in river flow due to dams built in Turkey and Syria, which negatively affects agriculture and hydroelectric energy production. The Euphrates River in Anbar suffers from pollution due to industrial and agricultural sewage, which affects the quality of the river's water and the population's health [22]. The Euphrates River is suffering from a decrease in discharge due to climate change and the upstream countries [23]. The river witnessed a decline in its water revenues by up to 30% after Iraq's neighboring countries began implementing development projects and building dams at the river's sources, especially in Turkey. As a result, the annual yield of the Euphrates River

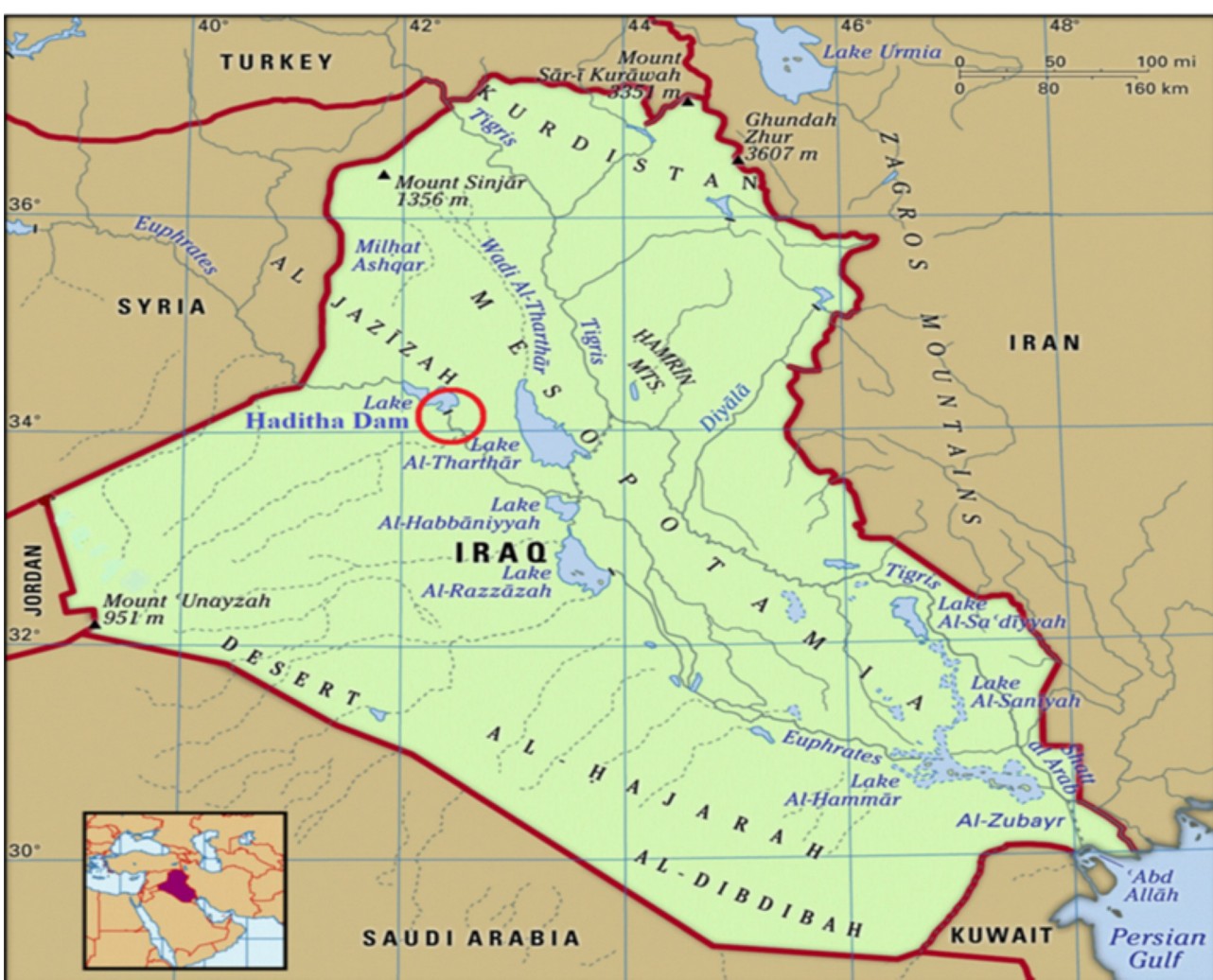

**Fig 1. The red circle indicates the location of the study area, the Haditha Dam reservoir location in Iraq [21].**

decreased from 30 billion cubic meters annually to 17 billion cubic meters annually. If the upstream countries continue to implement these projects and complete all plans, revenues reaching Iraq will gradually decrease to 24% by 2035; this indicates that Iraq will be significantly affected by these development projects and dams built on the Euphrates River, which will affect water availability and its future needs. Due to its importance in light of these conditions to which Iraq is exposed in terms of drought, the Euphrates River was chosen, which constitutes a large part of the water sources in Iraq, precisely the provider of the Haditha Dam reservoir, which is considered the only dam on this river, and all Iraqi cities derive their municipal and industrial water and agriculture from the water of this river.

## Data used

The study relied on the discharges of the Euphrates River at the Husaybah hydrological station, one of the main old stations in Iraq located on the river in Al-Qaim town, west of Anbar Governorate. Time series data on daily discharges from the Haditha Dam Project Administration - the General Authority for Dams and Reservoirs of the Iraqi Ministry of Water Resources

**Table 1. Statistical characteristics of drainage data for the study area.**

| Statistics | | Years | Max. | Min. | Mean | S.D |
|---|---|---|---|---|---|---|
| Daily (m³/sec) | Whole data set | 1985-2024 | 3361 | 50 | 551.2 | 369.25 |
| | Training & Validation data set | 1985-2012 | 3361 | 75 | 604.02 | 405.35 |
| | Testing data set | 2012-2024 | 1600 | 50 | 432.25 | 230.06 |
| Monthly (m³/sec) | Whole data set | 1985-2024 | 2984 | 97 | 551.08 | 347.1 |
| | Training & Validation data set | 1985-2012 | 2984 | 159 | 607.15 | 378.94 |
| | Testing data set | 2012-2024 | 964 | 97 | 422.3 | 207.22 |
| Seasonal (m³/sec) | Whole data set | 1985-2024 | 3459 | 105 | 551.25 | 319.52 |
| | Training & Validation data set | 1985-2012 | 2459 | 214 | 607.44 | 346.2 |
| | Testing data set | 2012-2024 | 897 | 105 | 426.48 | 196.18 |

(unpublished data) were collected from 1985 - 2024. It was observed that the highest value of discharges was (3361) m³/s in 1988, while the lowest was (50) m³/s in 2015. Daily time series data of the river was used, then converted to monthly and seasonal data. Table 1 shows the statistical characteristics of the river's discharge series, while (Fig 2) shows the time series of daily river discharges.

## Support Vector Machine (SVM)

Support vector machine (SVM) refers to supervised learning techniques that examine data and identify patterns for regression analysis and classification. The SVM learning system uses a hypothesis space of linear functions in a high-dimensional feature space. It is taught using an optimization theory-derived learning algorithm that applies a learning bias from statistical learning theory [24]. Vapnik [25] presented this learning technique as implementing the structural risk reduction concept. The hyperplane level and Lagrange multipliers, two crucial factors that significantly impact classification accuracy, are the foundation of the Support Vector Machine (SVM) model. The underlying data is represented in the input space, where the classification process is carried out. A hyperplane is defined as a boundary between different

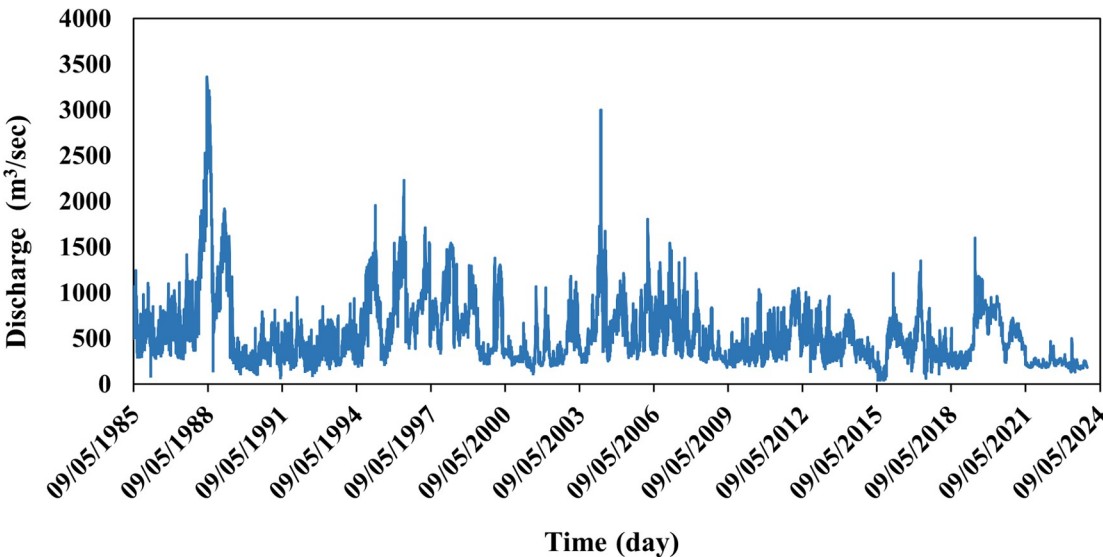

**Fig 2. Time series of daily discharges of the Euphrates River.**

categories or groups in the data. This level is determined based on the characteristics and information extracted from the training data. In addition, support machine models (SVMs) use Lagrange multipliers to achieve optimal cutoffs between classes. These multipliers are adjusted based on data characteristics and desired performance expectations. The support vector machine (SVM) effectively deals with various problems, including classifying data in a nonlinear space. This technique defines cutoffs in a way that minimizes classification error and enhances the overall accuracy of the analytical model. An appropriate linear separator that best separates the data into two classes is found through the SVM algorithm. The separator is the super level that maximizes the margin between the supporting data and the remaining errors. The margin is defined as the distance between the hyperplane and the nearest data points, and these points are called support points. SVM is used in wide applications in artificial intelligence, such as future predictions of flood discharges of rivers evaporation from lakes. This technique has achieved great success in many fields. However, SVM must be trained and parameterized with an extensive data set to achieve satisfactory performance. The SVM model requires careful selection of parameters such as kernel type and setting of parameter values. It may be difficult to specify these parameters appropriately, and trial and repeated adjustment may be required to obtain optimal performance. The SVM model requires careful selection of parameters such as kernel type and setting of parameter values. It may be difficult to specify these parameters appropriately, and trial and repeated adjustment may be required to obtain optimal performance. When dealing with large datasets, SVM model training can be expensive in terms of time and resources. Getting quick results may take time in these cases.

Consider regression within the collection of linear functions that reflect the data. Eq (1) [26].

$$f(x) = w^T x + b \tag{1}$$

N is the number of training values, while input $x_k \in R^n$ and $y_k \in R$ are the output values.

The following initial problem may then be used to define the optimization problem.

$$Min\, J(w, \xi, \xi*) = \frac{1}{2} w^T w + c \sum_{K=1}^{N} \left( \xi k + \xi_k^* \right) \tag{2}$$

$$Such\ that \begin{cases} y_k - w^T x_k - b \leq \varepsilon + \xi_k \\ w^T x_k + b - y_k \leq \varepsilon + \xi_k^* \\ \xi_k, \xi_k^* \geq 0 \end{cases} \tag{3}$$

The constant c determines the tolerance level for deviations from the desired $\varepsilon$ accuracy, and it is associated with slack variables $\xi_k$ and $\xi_k^*$ for k = 1, . . ., N. The issue must first be expressed in Lagrange form, after which the dual problem's quadratic programming must solve it. The linear function is converted in the double space.

$$f(x) = \sum_{k=1}^{N} (\alpha_k + \alpha_k^*) x_k^T x + b \tag{4}$$

With $\sum_{k=1}^{N}(\alpha_k + \alpha_k^*)x_k$ and $\alpha_k + \alpha_k^*$ are the Lagrange multipliers. The fundamental weight space model that follows is considered to facilitate SVM predictions for a nonlinear scenario.

$$f(x) = w^T \phi(x) + b \tag{5}$$

Applying the mapping $\phi(x): R^n \to R^{nh}$ to a high dimensional feature space. The kernel methods have been used in that case, resulting in K $(x_k, x_l) = \phi(x_k)^T \phi(x_l)$ for k = 1,. . ., N.

**Table 2. Tuning components for three different kernels in SVR.**

| Type of kernel functions | Tuning or affecting parameters |
|---|---|
| Linear | C |
| polynomial kernel | C and $\gamma$ |
| Gausses (RBF) | C, $\gamma$ and r |

Different kernel functions were used in designing the SVM with type space ε. The support model incorporates additional kernel functions such as the polynomial kernel, linear kernel function, and radial basis functions (RBFs) [27], as in Table 2, which were used in this study due to their popularity and wide use. It is worth noting that the vector machine calculations were based on programming in MATLAB, and the parameters were optimized.

$$K\left(xi, xj\right) = \left(\gamma\, X_i^T \times X_j + r\right)^d \tag{6}$$

$$K\left(xi, xj\right) = exp\left(-\frac{||xi - x||}{\sigma}\right) \tag{7}$$

$$K\left(xi, x\right) = X_i^T \times X_j \tag{8}$$

## Statistical measurements

**Determination coefficient ($R^2$).** Determination coefficient is a measure to evaluate how well a prediction model fits the observed data. It is measures the extent to which the model can explain the variance in observed data. The formula for the coefficient of determination is shown in Eq (9) [28].

$$R^2 = \left[\frac{\sum_{i=1}^n (Q_{obs} - \bar{Q}_{obs})(Q_{pre} - \bar{Q}_{pre})}{\sum_{i=1}^n \left(Q_{obs} - \bar{Q}_{obs}\right)^2 * \sum_{i=1}^n \left(Q_{pre} - \bar{Q}_{pre}\right)^2}\right]^2 \tag{9}$$

**Root Mean Square Error (RMSE).** Root Mean Square Error is a measure used to evaluate the accuracy of a data prediction. It is an improvement of the standard square error (MSE) as it takes the square root of the MSE value to bring it into the same unit of measure as the original data. The formula for RMSE is shown in Eq (11) [29].

$$RMSE = \sqrt{\frac{1}{n}\sum_{i=1}^n \left(Q_{obs} - Q_{pre}\right)^2} \tag{10}$$

**Mean Absolute Error (MAE).** Mean Absolute Error is a measure used to evaluate the accuracy of a prediction or prediction model. It measures the average of the absolute errors between the predicted values and the actual values in the data set, the formula for MAE is as shown in Eq (11) [30].

$$MAE = \frac{\sum_{i=1}^n |Q_{obs} - Q_{pre}|}{n} \tag{11}$$

Where,

$Q_{obs}$&$Q_{pre}$: Value of the observation and predicted discharge data, respectively.

**Table 3. Different types of inputs and outputs in SVM.**

| Model | Input combination | Output Variable |
|-------|-------------------|-----------------|
| Model-1 | $Q_{t-1}$ | $Q_t$ |
| Model-2 | $Q_{t-1}.Q_{t-2}$ | $Q_t$ |
| Model-3 | $Q_{t-1}.Q_{t-2}.Q_{t-3}$ | $Q_t$ |
| Model-4 | $Q_{t-1}.Q_{t-2}.Q_{t-3}.Q_{t-4}$ | $Q_t$ |
| Model-5 | $Q_{t-1}.Q_{t-2}.Q_{t-3}.Q_{t-4}.Q_{t-5}$ | $Q_t$ |

$Q^-_{obs}$ & $Q^-_{pre}$: Mean values of observation and prediction discharge data, respectively.

n = number of real data.

## Training process

Different training patterns were adopted using the previous values as input to predict the later values, as in Table 3 below. While the (Fig 3) shows the flowchart of the mechanism for predicting future discharges of the Euphrates River upstream of the Haditha Dam reservoir and how to divide, train and test the data using the SVM model.

## Results and discussion

Before the training process, the data was divided into two groups. The first group is the training and calibration group, representing 70% of the river discharge data. The second set is the test set, representing 30% of the river discharge data.

The SVR technique was applied in MATLAB three-time horizons (daily, monthly and seasonal) with different kernels: linear, quadratic, Gaussian or RBF. These kernels were used to find the most accurate kernel. Five models with varying input sets were applied to the three different time horizons, as shown in Table 3. This process aims to study the effect of the response of daily discharges to previous values in predicting subsequent discharges. After completing the training process, the test data was used to make predictions and measure the capacity of time delay prediction; this way, the models can predict future values based on past values. This process aims to analyze and evaluate the relationship between the river's daily, seasonal and monthly discharges and the impact of previous values on forecasting and better understanding the behavior of the water drainage system. SVR models are developed and compared regarding RMSE and $R^2$, with different kernel functions and designed input parameters. A model that produces lower errors will reflect higher performance in this prediction of reservoir flow. Different kernel parameters were used as tuning parameters to improve the model accuracy. Several tuning or effect parameters are used in the SVR kernel. (Fig 4A) show the results of training the model on daily discharge rates and comparing them with the observed values observed for the same period. (Fig 5A) show the relationship between the observed values and the predicted values for the linear kernel function training phase with a determination factor of $R^2 = 0.96$. (Fig 4B) show the results of training the model on monthly predicted rates and comparing them with the observed values for the same period. (Fig 5B) show the relationship between the observed values and the predicted values for the linear kernel function training phase with a determination factor of $R^2. = 0.68$. Also, (Fig 4C) show the results of training the model on seasonal discharge rates and comparing them with the observed values for the same period. (Fig 5C) show the relationship between the recorded observed values and the predicted values for the training phase of the quadratic kernel function with a determination factor of $R^2 = 0.21$.

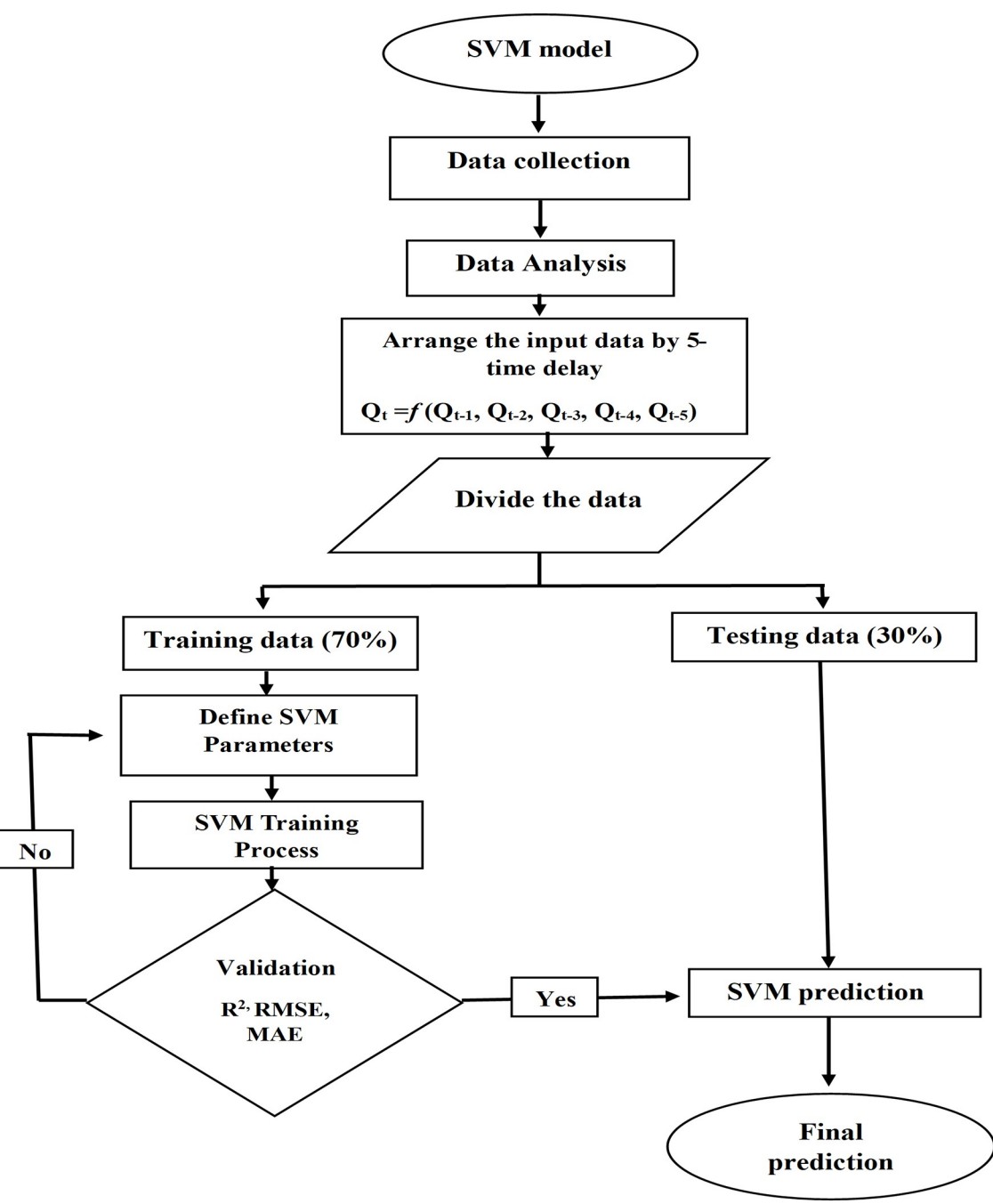

**Fig 3. Flowchart of the prediction mechanism using the SVM model.**

Table 4 shows the results obtained from training the models and verifying their performance on the daily river discharges based on several successive time delays shown in Table 3. After analyzing the results, it was concluded that the linear kernel outperformed the other kernels (Quadratic and Gaussian) in predicting the daily data rate using a one-day time delay. This superiority was measured using significant statistical performance measures, namely the coefficient of determination ($R^2$), the root mean square error (RMSE) and mean absolute error (MAE). The value of ($R^2$) for the linear kernel in the testing phase was equal to (0.95), as in the

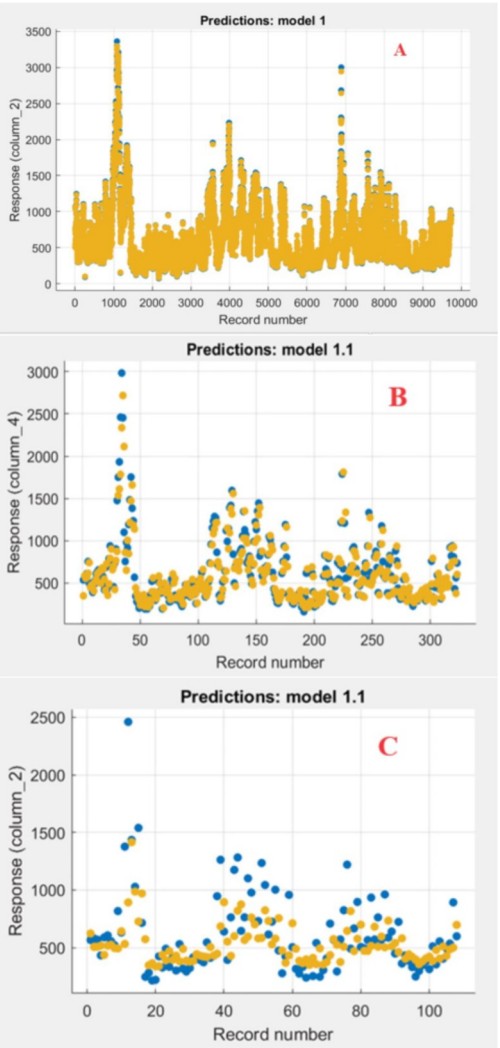

**Fig 4.** Training process of the SVM model for discharges (a) daily, (b) monthly, and (c) seasonal.

(Fig 6A), which means that it achieved the highest ability to agree between predicted values and observed values compared to other kernels, as in the (Fig 7A). The value of (RMSE) in the testing phase was equal to (53.29) m$^3$/sec, the lowest value among the compared kernels, indicating higher prediction accuracy and less deviation between the predicted values and the observed values. While the MAE was Value equal to (33.26) m$^3$/sec. Based on these results, it can be concluded that the linear kernel outperforms other kernels in using a 1-day time delay to predict the daily data rate.

While Table 5 shows the results obtained from training the models and verifying their performance on the monthly river discharges based on several successive time delays shown in Table 3, after analyzing the results, it is noted that the performance of the linear kernel is also superior to the other kernels, with a time delay of three days in predicting the monthly rate of discharges through comparison with the statistical coefficients ($R^2$), (RMSE) and (MAE). The value of ($R^2$) in the testing phase was equal to (0.731) as in (Fig 6B), which is the best value compared to the other cores. In contrast, the value of (RMSE) in the testing phase is equal to (109.4) m$^3$/sec, which is the lowest value and is considered the best among them, which means

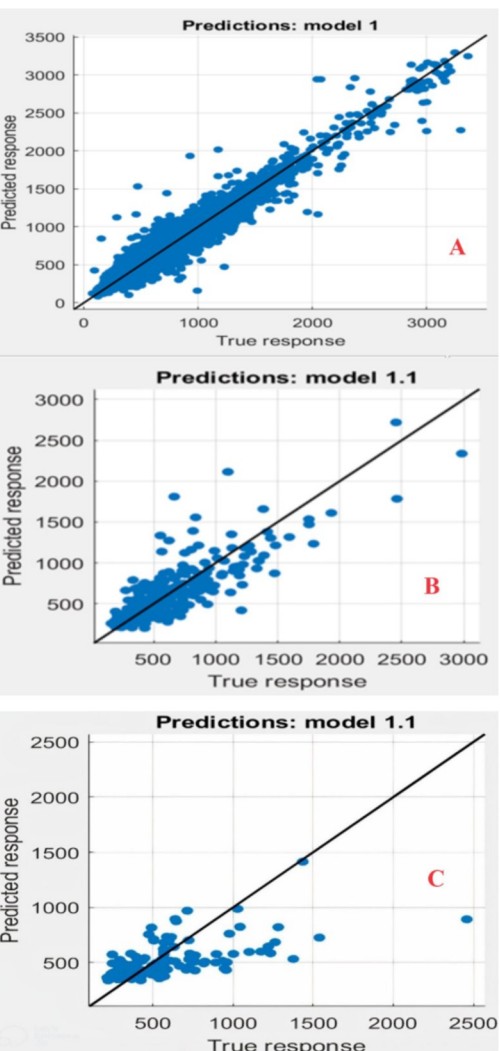

**Fig 5.** $R^2$-value of observed versus predicted flow for the training phase SVM model for discharges (a) daily, (b) monthly, and (c) seasonal.

that it achieved the highest ability to agree between predicted values and observed values compared to other kernels, as in the (Fig 7B). While the MAE was Value equal to (76.08) m³/sec. Based on these results, the linear kernel also performs better than other kernels when using a three-day time delay in predicting a monthly data rate.

While Table 6 shows the results obtained from training the models and verifying their performance on the seasonal river discharges based on several successive time delays shown in Table 3, after analyzing the results, it is noted that the quadratic kernel is superior to the other kernels in prediction average the data seasonal using a time delay of 1 day. The comparison between the quadratic kernel and other kernels was done using well-known statistical performance metrics, namely the coefficient of ($R^2$), (RMSE) and (MAE). The quadratic kernel showed a value ($R^2$) equal to (0.415) in the testing phase, as shown in (Fig 6C), which indicates the strength of agreement between the predicted values and the observed values better than other kernels, as shown in (Fig 7C). In addition, the value of (RMSE) for the linear kernel was estimated at (152.3) m³/sec, the lowest value, indicating high prediction accuracy and low

**Table 4. Indicators for evaluating daily discharge performance of SVM kernel functions.**

| Model | | Train | | | Test | | |
|---|---|---|---|---|---|---|---|
| | | $R^2$ | RMSE | MAE | $R^2$ | RMSE | MAE |
| Linear | Model-1 | 0.96 | 88.07 | 54.39 | 0.95 | 53.29 | 33.26 |
| | Model-2 | 0.95 | 88.33 | 54.09 | 0.945 | 54.05 | 33.86 |
| | Model-3 | 0.95 | 87.55 | 53.37 | 0.945 | 54.2 | 33.66 |
| | Model-4 | 0.95 | 87.41 | 53.35 | 0.945 | 53.83 | 33.49 |
| | Model-5 | 0.95 | 87.01 | 53.08 | 0.945 | 54.01 | 33.54 |
| Quadratic | Model-1 | 0.94 | 97.9 | 66.65 | 0.9465 | 58.41 | 42.7 |
| | Model-2 | 0.95 | 90.3 | 57.35 | 0.9444 | 64.21 | 49.3 |
| | Model-3 | 0.95 | 87.9 | 55.12 | 0.9436 | 56.65 | 38.22 |
| | Model-4 | 0.95 | 91.7 | 59.32 | 0.944 | 57.41 | 39.97 |
| | Model-5 | 0.95 | 91.7 | 59.28 | 0.9431 | 55.08 | 34.52 |
| Fine Gaussian | Model-1 | 0.95 | 90.78 | 55.98 | 0.931 | 61.1 | 37.7 |
| | Model-2 | 0.94 | 102.07 | 58.21 | 0.921 | 65.5 | 39.01 |
| | Model-3 | 0.92 | 114.54 | 60.41 | 0.913 | 68.8 | 40.41 |
| | Model-4 | 0.9 | 128.74 | 63.64 | 0.913 | 69 | 40.78 |
| | Model-5 | 0.88 | 139.61 | 66.52 | 0.914 | 68.7 | 41.04 |
| Medium Gaussian | Model-1 | 0.95 | 88.47 | 55.05 | 0.947 | 53.45 | 35.05 |
| | Model-2 | 0.95 | 88.55 | 54.97 | 0.945 | 54.79 | 36.51 |
| | Model-3 | 0.95 | 88.92 | 54.32 | 0.944 | 54.79 | 35.1 |
| | Model-4 | 0.95 | 89.33 | 54.38 | 0.944 | 54.73 | 35.78 |
| | Model-5 | 0.95 | 89.72 | 54.52 | 0.944 | 54.89 | 35.78 |
| Coarse Gaussian | Model-1 | 0.95 | 88.16 | 55.28 | 0.946 | 53.67 | 35.61 |
| | Model-2 | 0.95 | 87.95 | 55.26 | 0.944 | 54.87 | 36.94 |
| | Model-3 | 0.95 | 87.05 | 54.43 | 0.944 | 54.68 | 36.52 |
| | Model-4 | 0.95 | 87.1 | 54.62 | 0.945 | 53.96 | 35.64 |
| | Model-5 | 0.95 | 86.8 | 54.58 | 0.945 | 54.25 | 36.37 |

deviation between the predicted and observed values. While the MAE was Value equal to (129.3) m³/sec. Based on these results, the quadratic kernel performs better than other kernels in predicting seasonal data rates using a time delay of 1 day. The (Fig 8) shows the difference in the value of MAE between the daily, monthly and seasonal discharges in the testing phase of the models used. It is noted that the lowest value of the statistical coefficient MAE was on the daily discharges of the first model using the linear kernel function, where its value reached (33.26) m³/sec, which is the lowest value compared to the monthly and seasonal discharges. The same applies to the other kernels, which are Gaussian and quadratic. While the (Fig 9) shows the difference in the value of RMSE between the daily, monthly and seasonal discharges in the testing phase of the models used. It is noted that the lowest value of the statistical coefficient RMSE was on the daily discharges of the first model using the linear kernel function, where its value reached (53.29) m³/sec, which is the lowest value compared to the monthly and seasonal discharges. The same applies to the other kernels, which are Gaussian and quadratic. Table 7 shows the statistical characteristics between the observed and predicted values for each of the daily, monthly and seasonal discharge. By comparing the results, it is noted that the observed and predicted values for the daily discharge are very good. As for the monthly and seasonal discharge, there is a closeness but less than the daily discharge.

Based on the results obtained, as shown in Tables 4–6 above, it is clear that using a linear kernel can provide excellent performance in predicting the daily discharge of the Euphrates

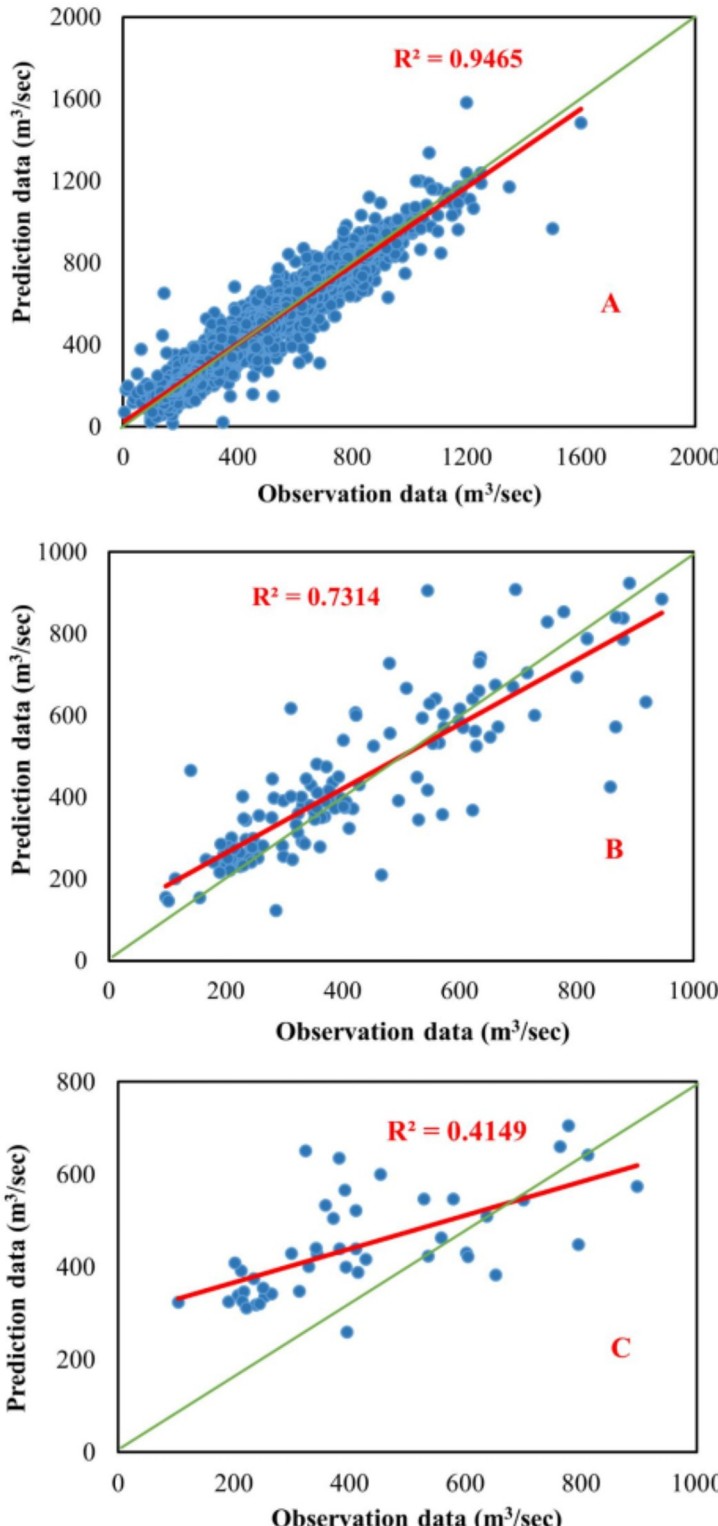

**Fig 6.** $R^2$ value of observed versus predicted flow for the testing phase SVM model for discharges (a) daily, (b) monthly, and (c) seasonal.

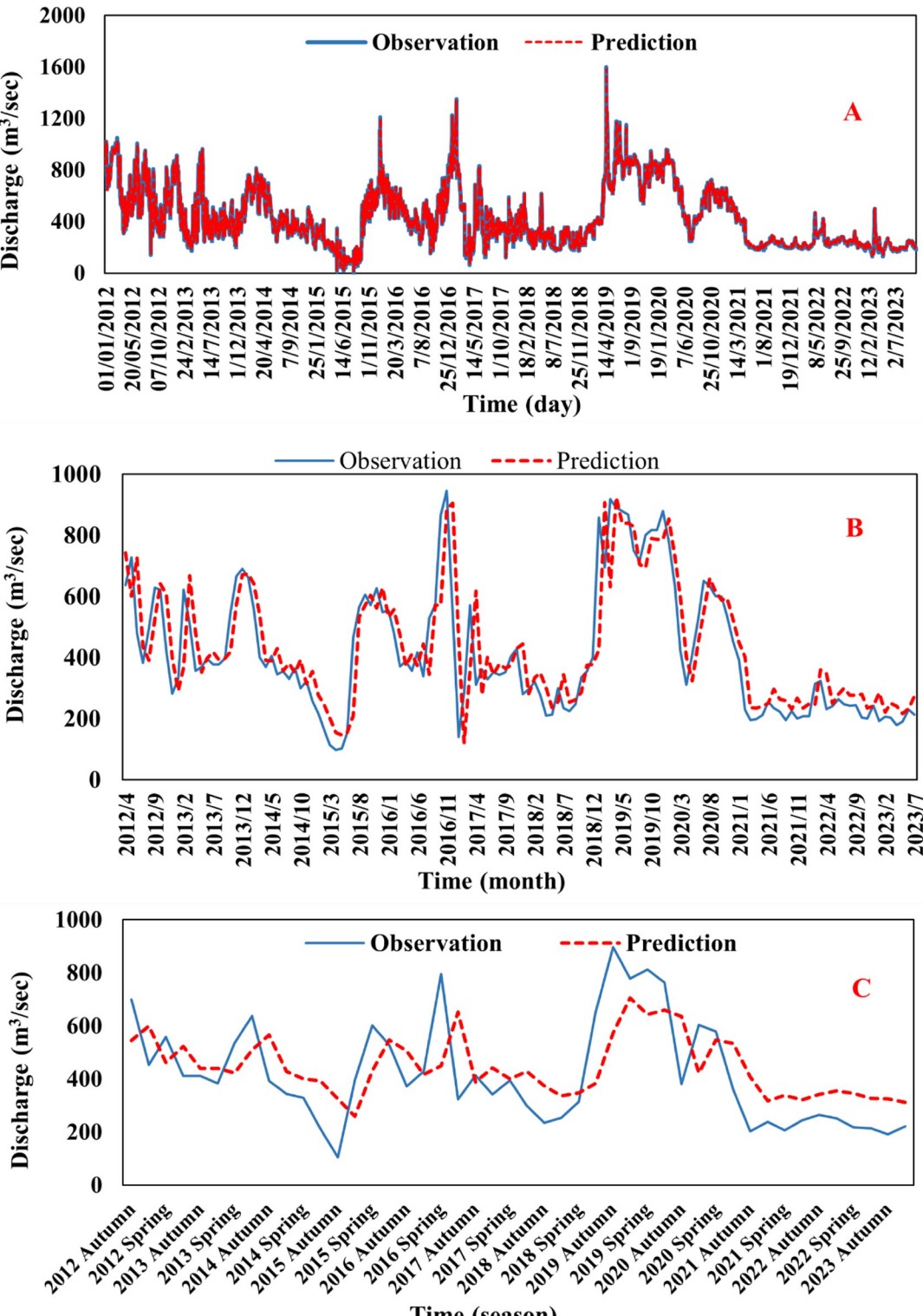

**Fig 7.** Time series of observation and predicted discharges for the model SVR on discharges (a) daily, (b) monthly, and (c) seasonal.

**Table 5. Indicators for evaluating monthly discharge performance of SVM kernel functions.**

| Model | | Train | | | Test | | |
|---|---|---|---|---|---|---|---|
| | | $R^2$ | RMSE | MAE | $R^2$ | RMSE | MAE |
| Linear | Model-1 | 0.69 | 214.37 | 147.25 | 0.718 | 111.3 | 76.23 |
| | Model-2 | 0.69 | 212.71 | 143.87 | 0.73 | 110.5 | 78.45 |
| | Model-3 | 0.68 | 213.94 | 144.87 | 0.731 | 109.4 | 76.08 |
| | Model-4 | 0.68 | 215.32 | 145.43 | 0.729 | 109.8 | 76.61 |
| | Model-5 | 0.68 | 214.58 | 146.46 | 0.727 | 110.2 | 77.76 |
| Quadratic | Model-1 | 0.86 | 216.14 | 148.81 | 0.72 | 111.3 | 76.23 |
| | Model-2 | 0.67 | 219.22 | 147.29 | 0.721 | 113.8 | 76.17 |
| | Model-3 | 0.55 | 254.39 | 160.51 | 0.727 | 111.9 | 72.68 |
| | Model-4 | 0.54 | 256.54 | 163.09 | 0.725 | 111.9 | 72.97 |
| | Model-5 | 0.48 | 273.19 | 166.32 | 0.713 | 115.4 | 75.99 |
| Fine Gaussian | Model-1 | 0.54 | 257.91 | 158.74 | 0.562 | 141.4 | 93.82 |
| | Model-2 | 0.45 | 284.05 | 169.39 | 0.471 | 157 | 109.9 |
| | Model-3 | 0.37 | 301.12 | 183.3 | 0.553 | 148 | 105.86 |
| | Model-4 | 0.35 | 505.16 | 184.55 | 0.482 | 155 | 113.2 |
| | Model-5 | 0.32 | 313.12 | 191.14 | 0.489 | 159 | 119.46 |
| Medium Gaussian | Model-1 | 0.58 | 247.54 | 155.34 | 0.718 | 112.5 | 80.43 |
| | Model-2 | 0.55 | 255.83 | 159.34 | 0.691 | 119.4 | 89.19 |
| | Model-3 | 0.56 | 249.8 | 154.15 | 0.699 | 118.6 | 89.48 |
| | Model-4 | 0.58 | 245.38 | 151.38 | 0.693 | 118.3 | 87.36 |
| | Model-5 | 0.52 | 264.87 | 159.53 | 0.702 | 115.9 | 86.25 |
| Coarse Gaussian | Model-1 | 0.62 | 236.61 | 152.82 | 0.714 | 112.45 | 80.43 |
| | Model-2 | 0.63 | 231.77 | 152.97 | 0.721 | 112.17 | 83.06 |
| | Model-3 | 0.63 | 229.66 | 152.52 | 0.716 | 112.7 | 82.16 |
| | Model-4 | 0.63 | 231.51 | 156.68 | 0.711 | 113.3 | 82.74 |
| | Model-5 | 0.61 | 237.25 | 158.12 | 0.698 | 114.9 | 83.61 |

River upstream of Haditha reservoir, especially when a time delay of one day is applied. That confirms that the linear kernel shows superior ability in predicting discharge rates with high accuracy and reliability, indicating substantial agreement between the predicted and observed values. In addition, the coefficient of determination ($R^2$) and root mean square error (RMSE) values suggest that the linear kernel provided remarkably accurate estimates, with a correlation coefficient of 0.95 and a root mean square error of 53.29, the lowest values recorded. Also, using a linear kernel can provide excellent performance in predicting the monthly discharge of the Euphrates River, especially when a three-day time delay is applied. The coefficient of determination ($R^2$) and root mean square error (RMSE) values indicate that the linear kernel provided remarkably accurate estimates, with the coefficient of determination of 0.731 and the root mean square error of 109.4 being the lowest values recorded. As for predicting seasonal discharge, using a quadratic kernel gives an acceptable performance in predicting monthly discharge, especially when applying a time delay of one day, based on the coefficient of determination, which reached (0.415) and the root mean square error (152.3).

Many global researches have used the same model for future prediction as [12, 16, 23, 28] was used on certain regions. In this study, the same model was used to predict the future daily, monthly, and seasonal discharges of the Euphrates River upstream of the Haditha Dam reservoir. After analyzing the results, we find that the model used has an acceptable performance in future prediction, and compared to the results of other research, there are no significant differences between the results. From observing the results obtained from applying the (SVM)

**Table 6. Indicators for evaluating seasonal discharge performance of SVM kernel functions.**

| Model | | Train | | | Test | | |
|---|---|---|---|---|---|---|---|
| | | $R^2$ | RMSE | MAE | $R^2$ | RMSE | MAE |
| Linear | Model-1 | 0.29 | 292.61 | 197.14 | 0.413 | 153.6 | 131.5 |
| | Model-2 | 0.26 | 299.13 | 201.19 | 0.379 | 154.5 | 129.5 |
| | Model-3 | 0.24 | 303.19 | 206.57 | 0.34 | 159 | 130.4 |
| | Model-4 | 0.23 | 303.24 | 196.83 | 0.222 | 176.1 | 140.8 |
| | Model-5 | 0.25 | 300.56 | 191.44 | 0.28 | 171.5 | 136.7 |
| Quadratic | Model-1 | 0.21 | 307.96 | 210.34 | 0.415 | 152.3 | 129.3 |
| | Model-2 | 0.41 | 311.6 | 170.96 | 0.3309 | 160.1 | 119.87 |
| | Model-3 | 0.28 | 340.05 | 178.26 | 0.3421 | 158.75 | 119.26 |
| | Model-4 | 0.14 | 321.54 | 205.46 | 0.112 | 198.3 | 159 |
| | Model-5 | 0.05 | 337.39 | 196.49 | 0.143 | 202.4 | 161.7 |
| Fine Gaussian | Model-1 | 0.18 | 314.15 | 216.95 | 0.322 | 175.3 | 149.8 |
| | Model-2 | 0.13 | 323.85 | 209.58 | 0.156 | 203.4 | 166.8 |
| | Model-3 | 0.07 | 335.04 | 226.83 | 0.253 | 188.9 | 163.2 |
| | Model-4 | 0.1 | 329.23 | 216.61 | 0.17 | 203.6 | 178.3 |
| | Model-5 | 0.12 | 325.27 | 206.18 | 0.055 | 227 | 196.2 |
| Medium Gaussian | Model-1 | 0.18 | 314.13 | 208.39 | 0.385 | 158.8 | 136.7 |
| | Model-2 | 0.13 | 325.15 | 211.59 | 0.265 | 170.6 | 139.2 |
| | Model-3 | 0.12 | 326.42 | 218.69 | 0.263 | 174.1 | 143.6 |
| | Model-4 | 0.17 | 316.2 | 202.75 | 0.122 | 195.2 | 162.2 |
| | Model-5 | 0.17 | 316.2 | 204.07 | 0.155 | 192.1 | 158.1 |
| Coarse Gaussian | Model-1 | 0.19 | 313.18 | 209.06 | 0.406 | 158.7 | 137.7 |
| | Model-2 | 0.17 | 316.85 | 205.06 | 0.338 | 162.6 | 140.3 |
| | Model-3 | 0.15 | 320 | 209.8 | 0.318 | 166.2 | 142.2 |
| | Model-4 | 0.21 | 307.81 | 199.73 | 0.21 | 176.1 | 146.1 |
| | Model-5 | 0.22 | 305.75 | 196.03 | 0.277 | 171.4 | 145.3 |

method on the Euphrates River on a daily basis and comparing them with previous studies such as [31–33] we notice that the values of statistical coefficients such as (MAE) were close or close to the general average of these studies, as well as the values of (RMSE) and (R2) When applied on a monthly or seasonal basis, the results were acceptable, but not as accurate as the daily forecast. The SVM model used to predict the daily discharges of the Euphrates River upstream of the Haditha Dam reservoir is considered acceptable and reliable and can provide many benefits for water resources management and the water sector in general. The model can predict the size and timing of potential floods; this can help develop flood coping strategies and improve early warning and risk management procedures. Forecasts can be used to determine future water needs and plan sustainable use of water resources; this helps determine adequate irrigation, storage, and groundwater management policies. Strategies can be developed to mitigate the effects of drought and improve water management in periods of drought. The allocation of water resources can be enhanced, precautionary measures can be applied, and water consumption can be controlled. Forecasts can be used to determine water infrastructure needs such as dams, canals, and other hydrological structures. Infrastructure planning and design can be improved, and more effective use of water resources can be achieved. With a better understanding of river discharges and their forecasts, strategies can be developed to protect and improve the aquatic environment. River management and environmental measures can be enhanced, and biodiversity and ecological balance can be maintained.

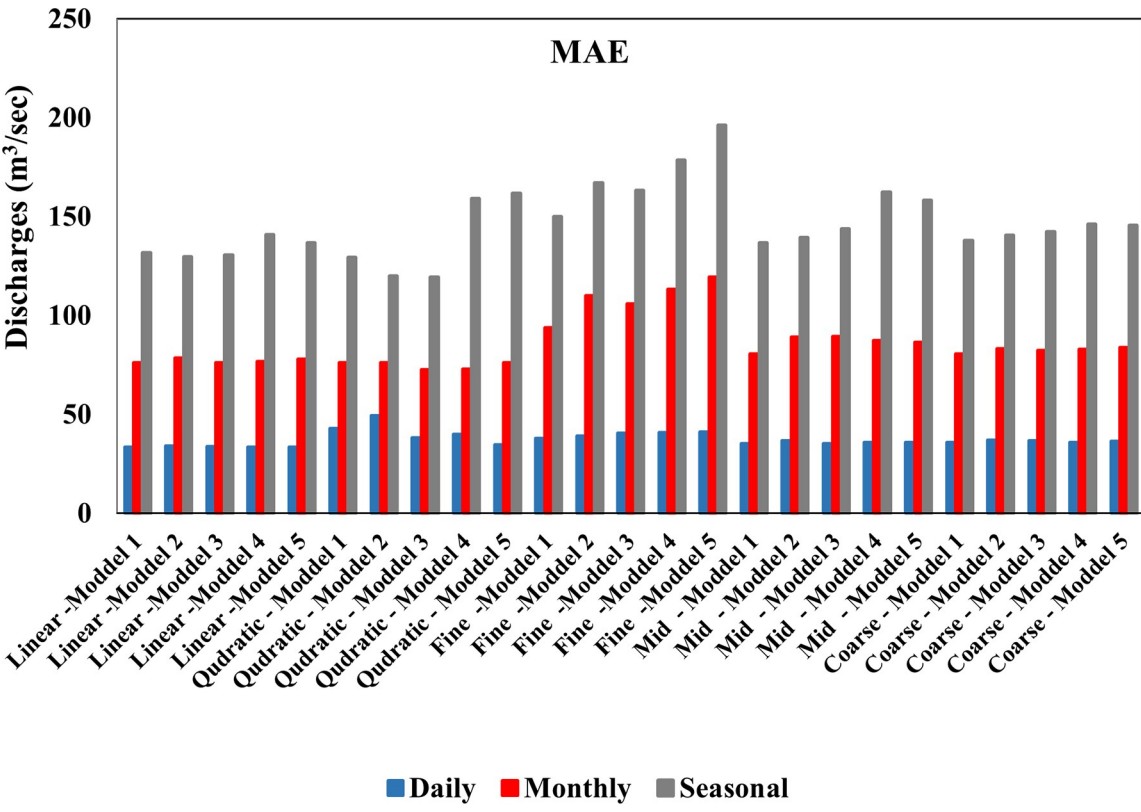

**Fig 8. AME value for the testing phase SVM model for daily, monthly, and seasonal discharges.**

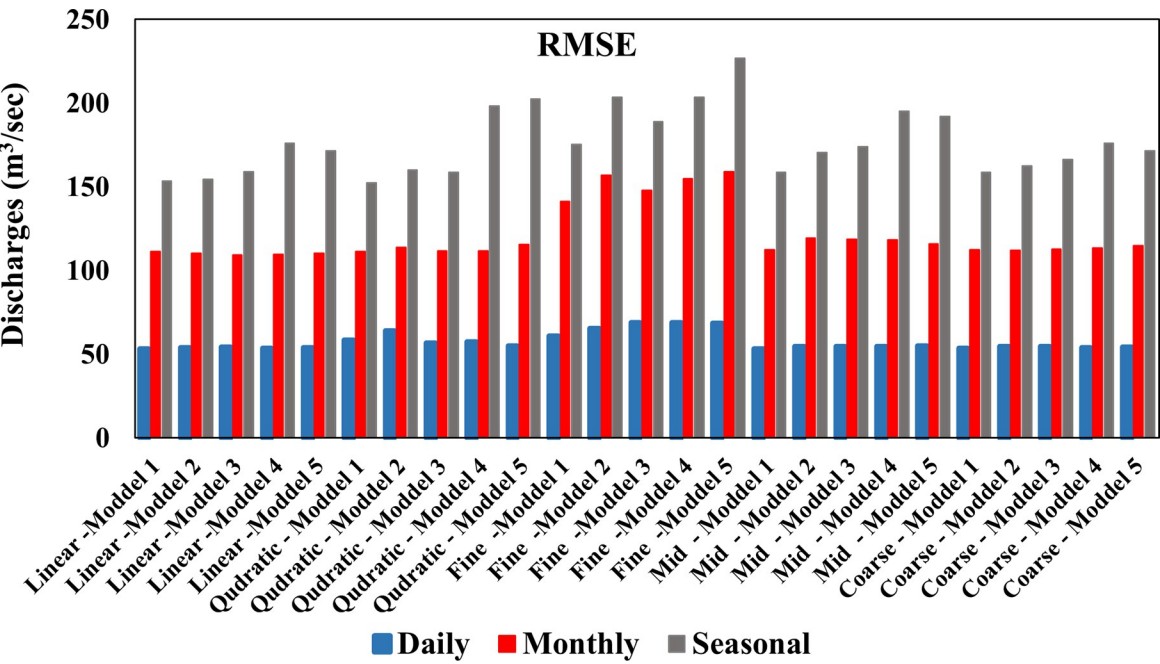

**Fig 9. RMSE value for the testing phase SVM model for daily, monthly, and seasonal discharges.**

**Table 7. Statistical characteristics between observation data and prediction data for the study area.**

| Statistic | Daily (m³/sec) | | Monthly (m³/sec) | | Seasonal (m³/sec) | |
|---|---|---|---|---|---|---|
| | Observation Data | Prediction Data | Observation Data | Prediction Data | Observation Data | Prediction Data |
| Max. | 1600 | 1582 | 964 | 924 | 897 | 706 |
| Min. | 50 | 15 | 97 | 123 | 105 | 260 |
| Mean | 432.25 | 434.49 | 422.3 | 434.94 | 426.48 | 446.24 |
| S.D | 230.06 | 226.16 | 207.22 | 190.65 | 196.18 | 110.59 |

## Conclusion and recommendations

The main conclusions of the present study could be summarized as follows:

• According to the results, the daily flow of the river obtained the highest accuracy compared to the seasonal and monthly time intervals.

• By comparing the statistical standards of the SVM models, we notice the superior performance of the linear kernel function to predict the daily discharge of the Euphrates River according to the coefficient of determination, which reached ($R^2 = 0.95$), which is the highest value compared to the other values, and (RMSE = 53.29), which is the lowest value, between them.

• Based on the results, the (SVR) model can be used to improve water resources management and dam operations for the Euphrates River upstream of the Haditha reservoir.

• The model (SVR) can support develop flood adaptation strategies, improve early warning and risk management measures, mitigate the effects of drought, and improve water management in drought periods within the selected area.

• A hybrid model is recommended to predict the Euphrates River discharges upstream of the Haditha Dam reservoir.

• Adopting time series data for other variables (rain, temperatures, evaporation coefficient, etc.) as inputs for forecasting models and investigating their impact on future discharges.

• It is recommended that SVM technology and artificial intelligence models be expanded to study various issues related to water resources in Iraq.

## Supporting information

**S1 Data. Haditha reservoir daily inflow 1985-2023.**
(XLSX)

## Acknowledgments

The authors would like to acknowledge their gratitude and appreciation to the College of Engineering, Department of Dams and Water Resources, Anbar University. Also, thanks and gratitude to the management of the Haditha Dam Project / Operating Authority for Dams and Reservoirs / Ministry of Water Resources in Iraq, who contributed the specific information in the research.

## Author Contributions

**Conceptualization:** Dhiya Al-Jumeily.

**Data curation:** Sadeq Oleiwi Sulaiman.

**Formal analysis:** Dhiya Al-Jumeily.

**Methodology:** Othman A. Mahmood.

**Project administration:** Sadeq Oleiwi Sulaiman, Dhiya Al-Jumeily.

**Software:** Othman A. Mahmood, Sadeq Oleiwi Sulaiman, Dhiya Al-Jumeily.

**Supervision:** Sadeq Oleiwi Sulaiman.

**Validation:** Othman A. Mahmood.

**Visualization:** Othman A. Mahmood.

**Writing – original draft:** Othman A. Mahmood.

**Writing – review & editing:** Othman A. Mahmood, Sadeq Oleiwi Sulaiman.

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
