## [Decision Letter · Decision Letter 0]

20 Mar 2024

PONE-D-24-07720Forecasting for Haditha Reservoir Inflow in the West of Iraq Using Support Vector MachinePLOS ONE

Dear Dr. Sulaiman,

Thank you for submitting your manuscript to PLOS ONE. After careful consideration, we feel that it has merit but does not fully meet PLOS ONE’s publication criteria as it currently stands. Therefore, we invite you to submit a revised version of the manuscript that addresses the points raised during the review process.

We look forward to receiving your revised manuscript.

Kind regards,

Sandeep Samantaray

Academic Editor

PLOS ONE

Journal Requirements:

4. We note that Figure 1 in your submission contain map/satellite images which may be copyrighted. All PLOS content is published under the Creative Commons Attribution License (CC BY 4.0), which means that the manuscript, images, and Supporting Information files will be freely available online, and any third party is permitted to access, download, copy, distribute, and use these materials in any way, even commercially, with proper attribution. For these reasons, we cannot publish previously copyrighted maps or satellite images created using proprietary data, such as Google software (Google Maps, Street View, and Earth). For more information, see our copyright guidelines: http://journals.plos.org/plosone/s/licenses-and-copyright.

5. Please ensure that you refer to Figures 1-4 in your text as, if accepted, production will need this reference to link the reader to the figure.

Additional Editor Comments:

1. Please modify the objective section for a clear understanding i.e novelty part should be clearly mentioned.

2. There are so many techniques in the recent world for the assessment; why does the author use a specified classical model for research purposes? Is there any specific reason for this? Suggest to add Hybrid model/ensable model.

3. Author must add statistical components/parameters of collected data in the study area section.

4. Equation 9-12; please add a recent citation for reference purposes. [Ex; 10.1016/j.jhydrol.2024.131042, 10.1016/j.jclepro.2024.141069, 10.1016/j.engappai.2023.107559]

5. Here author must be mentioned where they got the data and what is the span of the used data. Is there any specific reason for that?

6. Comparison statement (compare with other research articles) must be added in the result and discussion section to better visualize the proposed research.

7. Author must add future scope in the last portion of the manuscript.

8. Advantages and limitations of the proposed model must be added.

9. For better analysis of the result author must add a Histogram plot, box plot, and Taylor diagram

10. Author must provide a flow chart, pseudo code, and architecture of proposed models.

11. Author must provide a parameter table for clear understanding.

Reviewers' comments:

Reviewer's Responses to Questions

**Comments to the Author**

1. Is the manuscript technically sound, and do the data support the conclusions?

Reviewer #1: No

Reviewer #2: No

Reviewer #3: No

2. Has the statistical analysis been performed appropriately and rigorously? 

Reviewer #1: No

Reviewer #2: No

Reviewer #3: No

3. Have the authors made all data underlying the findings in their manuscript fully available?

Reviewer #1: No

Reviewer #2: No

Reviewer #3: No

4. Is the manuscript presented in an intelligible fashion and written in standard English?

Reviewer #1: No

Reviewer #2: No

Reviewer #3: No

5. Review Comments to the Author

Reviewer #1: I am glad that the authors effectively addressed my concerns and challenges in their research work. The authors' ability to provide timely and satisfactory responses to my queries reflects their strong commitment to adhering to scientific principles and conducting reliable research. This dedication benefits the scientific community and enhances our understanding of the subject matter. Therefore, based on the authors' satisfactory response, I find this version of the article to be acceptable.

Reviewer #2: Abstract is acceptable, however the authors should highlight the importance of their findings.

The introduction section should cover many recent studies. The authors should refer to studies related to application of AI in water resources management.

The objectives of the study should be clearly presented.

Fig. 1 The location of the study area should be improved.

Fig. 2 Time series of daily river discharges. If possible avoid using excel in plotting the figures.

Figure 3 and 4 should be under results section.

All the figures are generated from the software so should be presented in better way. The current way is not acceptable.

Conclusion needs to be revised and discuss the findings and also the limitations of the study.

Reviewer #3: In the submitted paper, the authors utilized a single model called SVR for forecasting inflow in the Haditha Reservoir. They employed two kernels, namely linear and Gaussian kernels, and concluded that the linear kernel performed better.

Here are my comments:

The significance of the paper is very limited.

I have come across several published papers in recent years that have employed more advanced models to forecast inflow in the same case study of the Haditha Reservoir. Therefore, your model is not novel, and there are already numerous papers addressing this topic [1]–[3].

[1] M. F. Allawi, I. R. Hussain, M. I. Salman, and A. El-Shafie, “Monthly inflow forecasting utilizing advanced artificial intelligence methods: a case study of Haditha Dam in Iraq,” Stoch. Environ. Res. Risk Assess., vol. 35, no. 11, pp. 2391–2410, 2021, doi: 10.1007/s00477-021-02052-7.

[2] M. M. Hameed, M. K. AlOmar, A. A. A. Al-Saadi, and M. A. AlSaadi, “Inflow forecasting using regularized extreme learning machine: Haditha reservoir chosen as case study,” Stoch. Environ. Res. Risk Assess., 2022, doi: 10.1007/s00477-022-02254-7.

[3] Z. Abd Saleh, “Forecasting by Box-Jenkins (ARIMA) Models to Inflow of Haditha Dam,” J. Babylon Univ. Eng. Sci., vol. 21, no. 5, pp. 1675–1685, 2013.

The literature review and introduction in the paper are weak.

The methodology has a potential flaw in that the comparison is limited to only one model, which may lead to misconceptions. When the objective is to achieve accurate predictions, it is important to consider a broader range of model cases rather than restricting the comparison to just a linear kernel and a radial basis kernel. This approach could create the impression of cherry-picking results to emphasize significant improvements derived from a particular model.

The research seems to focus solely on presenting statistical results without providing sufficient elaboration in interpreting those results.

Based on the data presented in the study, it is unclear what constructive contribution the researcher aims to make to the field of hydrology. The research only showcases the performance of the predictive model without offering any meaningful addition or contribution to the discipline of hydrology or water resources.

Considering the results presented in Table 3, it is questionable why only one input lag is used. In time series data, forecasting usually requires multiple input lags to capture the complex relationships between these lags and future inflow.

The paper utilizes non-meaningful statistical analysis parameters (see Table 3 and 4). For instance, RMSE and MSE are used, but it is known that RMSE is the square root of MSE. Hence, this information is redundant.

The statistical analysis is insufficiently robust.

There is a significant discrepancy in the results, as evident from the third and fourth figures.

There is no discussion section.

According to Figures 4 and 5, the best model is not suitable to forecast the majority of inflow values, particularly those that deviate significantly from the mean inflow. Also, this indicates that classical statistical assessments are inadequate.

The paper's organization and graphical figures are of low quality.

There is no validation assessment included.

The conclusion and recommendation section lacks strength.

6. PLOS authors have the option to publish the peer review history of their article (what does this mean?). If published, this will include your full peer review and any attached files.

Reviewer #1: No

Reviewer #2: No

Reviewer #3: No

---

## [Author Response · Author response to Decision Letter 0]

6 Apr 2024

PONE-D-24-07720

Forecasting for Haditha Reservoir Inflow in the West of Iraq Using Support Vector Machine (SVM)

PLOS ONE

1-3 Editor Comments

Author response: The authors greatly appreciate to the Editor’s comments.

All comments were responded to and implemented

1- Complete data were obtained from the Haditha Dam Project Administration - the General Authority for Dams and Reservoirs of the Iraqi Ministry of Water Resources (unpublished data) as described in the data used section.

2- The Euphrates River and the Haditha Dam reservoir are located between latitudes (34° 40' and 34° 13') north and longitudes (42° 26' and 41° 55') east. The highest flood water level for the reservoir and the river is at level 147 and covers about 500 km2 with 10 km of shoreline. Figure (1) shows the Euphrates River upstream of the Haditha Dam reservoir in Anbar Governorate, western Iraq [20].

3- Figures (1, 3) are referred to in the text as follows:

• Figure (1) shows the Euphrates River upstream of the Haditha Dam reservoir in Anbar Governorate, western Iraq [20].

• While (Fig 3) shows the flowchart of the mechanism for predicting future discharges of the Euphrates River upstream of the Haditha Dam reservoir and how to divide, train and test the data using the SVM model.

4. We note that Figure 1 in your submission contain map/satellite images which may be copyrighted. All PLOS content is published under the Creative Commons Attribution License (CC BY 4.0), which means that the manuscript, images, and Supporting Information files will be freely available online, and any third party is permitted to access, download, copy, distribute, and use these materials in any way, even commercially, with proper attribution. For these reasons, we cannot publish previously copyrighted maps or satellite images created using proprietary data, such as Google software (Google Maps, Street View, and Earth). For more information, see our copyright guidelines: http://journals.plos.org/plosone/s/licenses-and-copyright.

Author response: The authors greatly agree with the reviewer’s comment.

The appearance of the study area has changed

5. Please ensure that you refer to Figures 1-4 in your text as, if accepted, production will need this reference to link the reader to the figure.

Author response: The authors greatly agree with the reviewer’s comment.

The mentioned forms are referred to within the text of the speech

Additional Editor Comments:

1. Please modify the objective section for a clear understanding i.e novelty part should be clearly mentioned.

Author response: The authors greatly agree with the reviewer’s comment.

This study aims to use artificial intelligence with an SVM model to predict the discharge of the Euphrates River upstream of the Haditha Dam and improve water resources management and dam operations. The study uses historical data on the discharge of the Euphrates River upstream of Haditha Dam and information about flow behavior and the impact of the recently constructed dams in upstream countries. Also, this study aims to analyze and evaluate the relationship between the river's daily, monthly, and seasonal discharges and the effect of previous values on predicting and better understanding the behavior of the water drainage system.

2. There are so many techniques in the recent world for the assessment; why does the author use a specified classical model for research purposes? Is there any specific reason for this? Suggest to add Hybrid model/enable model.

Author response: The authors greatly agree with the reviewer’s comment.

A paragraph was added to the conclusions and recommendations section recommending using a hybrid model in the study area.

3. Author must add statistical components/parameters of collected data in the study area section.

Author response: The authors greatly agree with the reviewer’s comment.

Table (1) has been added with the data parameters used within the study area

4. Equation 9-12; please add a recent citation for reference purposes. [Ex; 10.1016/j.jhydrol.2024.131042, 10.1016/j.jclepro.2024.141069, 10.1016/j.engappai.2023.107559]

Author response: The authors greatly agree with the reviewer’s comment.

Adding the required citation to Equations has been added

5. Here author must be mentioned where they got the data and what is the span of the used data. Is there any specific reason for that?

Author response: The authors greatly agree with the reviewer’s comment.

The study relied on the discharges of the Euphrates River at the Husaybah hydrological station, one of the main old stations in Iraq located on the river in Al-Qaim town. Time series data of daily discharges were collected from 1985 to 2024.

6. Comparison statement (compare with other research articles) must be added in the result and discussion section to better visualize the proposed research.

Author response: The authors greatly agree with the reviewer’s comment.

Many global researches have used the same model for future prediction as [12-16-23-28] was used on certain regions. In this study, the same model was used to predict the future daily, monthly, and seasonal discharges of the Euphrates River upstream of the Haditha Dam reservoir. After analyzing the results, we find that the model used has an acceptable performance in future prediction, and compared to the results of other research, there are no significant differences between the results.

7. Author must add future scope in the last portion of the manuscript.

Author response: The authors greatly agree with the reviewer’s comment.

A paragraph has been added to the conclusions and recommendations section explaining Adopting time series data for other variables (rain, temperatures, evaporation coefficient, etc.) as inputs for forecasting models and investigating their impact on future discharges.

8. Advantages and limitations of the proposed model must be added.

Author response: The authors greatly agree with the reviewer’s comment.

The SVM model is used in classification and regression problems

The SVM learning system uses a hypothesis space of linear functions in a high-dimensional feature space.

The support vector machine (SVM) effectively deals with various problems, including classifying data in a nonlinear space.

SVM is used in wide applications in artificial intelligence, such as future predictions of flood discharges of rivers evaporation from lakes

The SVM model requires careful selection of parameters such as kernel type and setting of parameter values. It may be difficult to specify these parameters appropriately, and trial and repeated adjustment may be required to obtain optimal performance.

SVM model training can be expensive in terms of time and resources when dealing with very large datasets. Getting quick results may be difficult in these cases.

9. For better analysis of the result author must add a Histogram plot, box plot, and Taylor diagram

Author response: The authors greatly agree with the reviewer’s comment.

Figures have been added to compare the true and predicted values on daily, monthly, and seasonal discharges in the testing phases, as shown in the Results and Discussion section.

10. Author must provide a flow chart, pseudo code, and architecture of proposed models.

Author response: The authors greatly agree with the reviewer’s comment.

A flow chart has been added to the prediction mechanism using the SVM model, as shown in Figure (3) in the Training Process section.

11. Author must provide a parameter table for clear understanding.

Author response: The authors greatly agree with the reviewer’s comment.

Table 2 has been added showing the parameters and types of functions used

5. Review Comments to the Author

Reviewer #1: I am glad that the authors effectively addressed my concerns and challenges in their research work. The authors' ability to provide timely and satisfactory responses to my queries reflects their strong commitment to adhering to scientific principles and conducting reliable research. This dedication benefits the scientific community and enhances our understanding of the subject matter. Therefore, based on the authors' satisfactory response, I find this version of the article to be acceptable.

Author response: The authors greatly appreciate with the reviewer’s comment.

Reviewer #2: Abstract is acceptable; however, the authors should highlight the importance of their findings.

Author response: The authors greatly agree with the reviewer’s comment.

The results showed that the proposed machine learning model performed well in predicting the daily flow of the Euphrates River upstream of the Haditha Dam reservoir; this indicates that the model might effectively forecast flows, which helps improve water resource management and dam operations.

The introduction section should cover many recent studies. The authors should refer to studies related to application of AI in water resources management.

Author response: The authors greatly agree with the reviewer’s comment.

Resources related to artificial intelligence techniques in water resources management have been added as follows:

A study used the Muskingum model to predict floods in the United States of America and the United Kingdom. A combination of a hybrid of the bat algorithm (BA) and the particle swarm optimization (PSO) algorithm, i.e., the hybrid bat-swarm algorithm (HBSA) was used. The results showed that the Muskingum model represented by (HBSA) achieved excellent performance compared to other methods based on the squared deviations (SSD), the sum of the absolute deviations (SAD), the peak discharge error, and the time-to-peak error [13]. A study used the Adaptive Neuro-Fuzzy Inference System (ANFIS) model to study the case of the influence of climate on monthly flow in the Aydoughmoush basin in Iran for the period 1987 - 2007. The bat algorithm (BA), particle swarm optimization (PSO), and Genetic Algorithm (GA) were used to obtain the ANFIS parameter and obtain the best ANFIS structure. The results showed better climate index performance with six months' delays. The study indicated that ANFIS-BA obtained better results than ANFIS-PSO and ANFIS-GA, with a root mean square error (RMSE) of 25% and 30% less than ANFIS-PSO and ANFIS-GA, respectively [14]. In a study, was used two artificial intelligence models to predict subsurface evaporation rates, represented by the generalized neural network model for regression and the neural network for the radiative basis function. The model’s input variables for this model include temperature, wind speed, humidity, and water depth. The applied models utilize actual hydrological and climatological in an arid region in the Iraqi Western Desert for two soil types. The results showed that the neural network model (ANN) could accurately predict subsurface reservoir evaporation based on the correlation coefficient, which reached (0.936) for fine gravel soil and (0.959) for coarse gravel soil [16]. A study used a deep learning model to predict river courses on data from the Tigris River in Iraq. Two methods were used to collect samples: linear deep learning (LDL) and stratified deep learning (SDL) in deep learning algorithms. The results indicated that stratified deep learning (SDL) improves accuracy by approximately 7.96-94.6 concerning several evaluation criteria. Thus, it is worth noting that SDL outperforms (LDL) in monthly streamflow modelling [17].

The objectives of the study should be clearly presented.

 Author response: The authors greatly agree with the reviewer’s comment.

This study aims to use artificial intelligence with an SVM model to predict the discharge of the Euphrates River upstream of the Haditha Dam and improve water resources management and dam operations. The study uses historical data on the discharge of the Euphrates River upstream of Haditha Dam and information about flow behavior and the impact of the recently constructed dams in upstream countries. Also, this study aims to analyze and evaluate the relationship between the river's daily, monthly, and seasonal discharges and the effect of previous values on predicting and better understanding the behavior of the water drainage system.

Fig. 1 The location of the study area should be improved.

Author response: The authors greatly agree with the reviewer’s comment.

We have optimized the location of the study area as shown in Figure 1 in the Study Area section

Fig. 2 Time series of daily river discharges. If possible, avoid using excel in plotting the figures.

Author response: The authors greatly agree with the reviewer’s comment.

The format of the time series for daily discharges has been modified better, as in the data used section

Figure 3 and 4 should be under results section.

Author response: The authors greatly agree with the reviewer’s comment.

Figure 3 and Figure 4 have been moved to the Results section

All the figures are generated from the software so should be presented in better way. The current way is not acceptable.

Author response: The authors greatly agree with the reviewer’s comment.

All figures were presented in a better way, as shown in the results section

Conclusion needs to be revised and discuss the findings and also the limitations of the study.

Author response: The authors greatly agree with the reviewer’s comment.

The discussion of the results has been completely revised and as described in the Results and Discussion section

Reviewer #3: In the submitted paper, the authors utilized a single model called SVR for forecasting inflow in the Haditha Reservoir. They employed two kernels, namely linear and Gaussian kernels, and concluded that the linear kernel performed better.

Here are my comments:

The significance of the paper is very limited.

Author response: The authors greatly agree with the reviewer’s comment.

The artificial intelligence model represented by the support vector machine was used in future forecasting of the daily, monthly, and seasonal discharges of the Euphrates River upstream of the Haditha dam reservoir. The SVR model was used to predict future discharges. The proposed model contains four function kernels in forecasting, and the best of them were used: linear, quadratic, and Gaussian (RBF) kernels. There are three types of Gaussian kernel in the MATLAB format, and they were used, as shown in Tables 4. 5, and 6 in the Results and Discussion section. Therefore, the research focuses on more than just using two kernels. Instead, it focuses on the future prediction of river flows, and these two kernels were chosen as the best among the other kernels in forecasting the river's flow.

I have come across several published papers in recent years that have employed more advanced models to forecast inflow in the same case study of the Haditha Reservoir. Therefore, your model is not novel, and there are already numerous papers addressing this topic [1]–[3].

[1] M. F. Allawi, I. R. Hussain, M. I. Salman, and A. El-Shafie, “Monthly inflow forecasting utilizing advanced artificial intelligence methods: a case study of Haditha Dam in Iraq,” Stoch. Environ. Res. Risk Assess., vol. 35, no. 11, pp. 2391–2410, 2021, doi: 10.1007/s00477-021-02052-7.

[2] M. M. Hameed, M. K. AlOmar, A. A. A. Al-Saadi, and M. A. AlSaadi, “Inflow forecasting using regularized extreme learning machine: Haditha reservoir chosen as case study,” Stoch. Environ. Res. Risk Assess., 2022, doi: 10.1007/s00477-022-02254-7.

[3] Z. Abd Saleh, “Forecasting by Box-Jenkins (ARIMA) Models to Inflow of Haditha Dam,” J. Babylon Univ. Eng. Sci., vol. 21, no. 5, pp. 1675–1685, 2013.

Author response: The authors greatly agree with the reviewer’s comment.

Indeed, the model is not new, but due to its importance in light of these conditions to which Iraq is exposed in terms of drought, the Euphrates River was chosen, which constitutes a large part of the water sources in Iraq, precisely the provider of the Haditha Dam reservoir, which is consi

---

## [Editor Report · Decision Letter 1]

2 May 2024

PONE-D-24-07720R1Forecasting for Haditha Reservoir Inflow in the West of Iraq Using Support Vector Machine (SVM)PLOS ONE

Dear Dr. Sulaiman, 

Thank you for submitting your manuscript to PLOS ONE. After careful consideration, we feel that it has merit but does not fully meet PLOS ONE’s publication criteria as it currently stands. Therefore, we invite you to submit a revised version of the manuscript that addresses the points raised during the review process.**Kindly add more chart for better analysis ****Compare the result with previous study.**

We look forward to receiving your revised manuscript.

Kind regards,

Dr. Sandeep Samantaray

Academic Editor

PLOS ONE
---

## [Author Response · Author response to Decision Letter 1]

9 May 2024

Forecasting for Haditha Reservoir Inflow in the West of Iraq Using Support Vector Machine

PLOS ONE

 Comments

• Kindly add more chart for better analysis 

Author response: The authors greatly appreciate to the Editor’s comments.

The (Fig.8) shows the difference in the value of MAE between the daily, monthly and seasonal discharges in the testing phase of the models used. It is noted that the lowest value of the statistical coefficient MAE was on the daily discharges of the first model using the linear kernel function, where its value reached (33.26), which is the lowest value compared to the monthly and seasonal discharges. The same applies to the other kernels, which are Gaussian and quadratic. While the (Fig. 9) shows the difference in the value of RMSE between the daily, monthly and seasonal discharges in the testing phase of the models used. It is noted that the lowest value of the statistical coefficient RMSE was on the daily discharges of the first model using the linear kernel function, where its value reached (53.29), which is the lowest value compared to the monthly and seasonal discharges. The same applies to the other kernels, which are Gaussian and quadratic.

Fig 8. AME value for the testing phase SVM model for daily, monthly, and seasonal discharges

Fig 9. RMSE value for the testing phase SVM model for daily, monthly, and seasonal discharges

• Compare the result with previous study.

Author response: The authors greatly appreciate to the Editor’s comments.

Many global researches have used the same model for future prediction as [12-16-23-28] was used on certain regions. In this study, the same model was used to predict the future daily, monthly, and seasonal discharges of the Euphrates River upstream of the Haditha Dam reservoir. After analyzing the results, we find that the model used has an acceptable performance in future prediction, and compared to the results of other research, there are no significant differences between the results. From observing the results obtained from applying the (SVM) method on the Euphrates River on a daily basis and comparing them with previous studies such as [29][30][31] we notice that he values of statistical coefficients such as (MAE) were close or close to the general average of these studies, as well as the values of (RMSE) and (R2) When applied on a monthly or seasonal basis, the results were acceptable, but not as accurate as the daily forecast.

All comments were responded to and implemented

---

## [Editor Report · Decision Letter 2]

31 May 2024

PONE-D-24-07720R2Forecasting for Haditha Reservoir Inflow in the West of Iraq Using Support Vector Machine (SVM)PLOS ONE

Dear Dr. Sulaiman,

Thank you for submitting your manuscript to PLOS ONE. After careful consideration, we feel that it has merit but does not fully meet PLOS ONE’s publication criteria as it currently stands. Therefore, we invite you to submit a revised version of the manuscript that addresses the points raised during the review process.

1) Comments from PLOS Editorial Office: We note that one or more reviewers and the Academic Editor have recommended that you cite specific previously published works in an earlier round of revision. As always, we recommend that you please review and evaluate the requested works to determine whether they are relevant and should be cited. It is not a requirement to cite these works and you may remove them before the manuscript proceeds to publication. We appreciate your attention to this request

2) Please ensure that all figures are of sufficient quality for publication. We note that Fig 4 and Fig 5 need updating before potential publication, as these look like screenshots with poor resolution. Furthermore, Fig 1 appears to contain copyrighted material.

We look forward to receiving your revised manuscript.

Kind regards,

Hanna Landenmark

Staff Editor, PLOS ONE

on behalf of 

Sandeep Samantaray

Journal Requirements:

Additional Editor Comments:

Thank you for the revison.

---

## [Author Response · Author response to Decision Letter 2]

12 Jun 2024

Editor Comments

• Kindly add more chart for better analysis 

Author response: The authors greatly appreciate to the Editor’s comments.

1- All citations mentioned in the manuscript belong to my work and are part of the research methodology.

2- Regarding Figure (1), the issue related to copyright was solved by replacing the image of the study site with that of the source for Figure (1) and as shown in the manuscript in the Study Area section.

Figs 4. Training process of the SVM model for discharges (a) daily, (b) monthly, and (c) seasonal.

Fig 5. R2-value of observed versus predicted flow for the training phase SVM model for discharges (a) daily, (b) monthly, and (c) seasonal

Fig. 1 The red circle indicates the Location of the study area, the Haditha Dam reservoir location in Iraq [21].

All comments were responded to and implemented

---

## [Editor Report · Decision Letter 3]

22 Jul 2024

Forecasting for Haditha Reservoir Inflow in the West of Iraq Using Support Vector Machine (SVM)

PONE-D-24-07720R3

Dear Dr. %Sulaiman%,

We’re pleased to inform you that your manuscript has been judged scientifically suitable for publication and will be formally accepted for publication once it meets all outstanding technical requirements.

Kind regards,

Dr. Sandeep Samantaray

Academic Editor

PLOS ONE

Additional Editor Comments (optional):

Thank you for the revision. 
---

## [Editor Report · Acceptance letter]

29 Jul 2024

PONE-D-24-07720R3 

PLOS ONE

Dear Dr. Sulaiman, 

I'm pleased to inform you that your manuscript has been deemed suitable for publication in PLOS ONE. Congratulations! Your manuscript is now being handed over to our production team.

Kind regards, 

on behalf of

Dr. Sandeep Samantaray 

Academic Editor

PLOS ONE